

# Indirect flood impacts and cascade risk across interdependent linear infrastructures

Chiara Arrighi[1], Maria Pregnolato[2], Fabio Castelli[1]

[1]Department of Civil and Environmental Engineering, University of Firenze, Florence, Italy
[2]Department of Civil Engineering, University of Bristol, Bristol, UK

*Correspondence to*: Chiara Arrighi (chiara.arrighi@unifi.it)

**Abstract.** Floods are the most frequent and damaging natural threat worldwide. Whereas the assessment of direct impacts is well advanced, the evaluation of indirect impacts is less frequently achieved. Indirect impacts are not due to the physical contact with flood water but result from the reduced performance of infrastructures. Linear critical infrastructures (such as 10 roads and pipes) have an interconnected nature that may lead to failure propagation, so that impacts extend far beyond the inundated areas and/or period. This work presents the risk analysis of two linear infrastructure systems, i.e. the water distribution system (WSS) and the road network system. The evaluation of indirect flood impacts on the two networks is carried out for four flooding scenarios, obtained by a coupled 1D-quasi 2D hydraulic model. Two methods are used for assessing the impacts on the water distribution system and on the road network, a Pressure-Driven Demand network model 15 and a transport network disruption model respectively. The analysis is focused on the identification of: (i) common impact metrics; (ii) vulnerable elements exposed to the flood; (iii) similarities and differences of the methodological aspects for the two networks; (iv) risks due to systemic interdependency. The study presents an application to the metropolitan area of Florence (Italy). When interdependencies are accounted for, results showed that the risk to the WSS in terms of Population Equivalent (PE/year) can be reduced by 71.5% and 41.8%, if timely repairs to the WSS stations are accomplished by 60 and 20 120 minutes respectively; the risk to WSS in terms of pipes length (km/year) reduces by 53.1% and 15.6%. The study highlights that resilience is enhanced by system risk-informed planning, which ensures timely interventions on critical infrastructures; however, temporal and spatial scales are difficult to define for indirect impacts and cascade effects. Perspective research could further improve this work by applying a system-risk analysis to multiple urban infrastructures.

## 1 Introduction

Linear infrastructure systems such as Water Supply System (WSS), electricity and transportation are considered Critical Infrastructures (CIs) because their failure would jeopardize public health and economic security, with repercussions on the whole society (Fekete, 2019; Tarani et al., 2019; Lhomme et al., 2013). CIs are exposed to natural hazards, such as flooding; in particular, ~7.5% of road and rail infrastructures are exposed to a 1/100-year flood event worldwide (Koks et al. 2019). Flooding can damage CIs directly (when impacts are due to the physical contact with floodwaters, i.e. direct impacts) and 30 indirectly (when impacts are not due to the physical contact, and/or occur outside the inundated area in space or time, i.e. indirect or cascade impacts). Changes in socio-economic and climatic conditions, as well as infrastructure interdependencies, could aggravate both direct and indirect impacts in the future (Pregnolato et al., 2017a; Evans et al., 2020).

Existing studies offer well-established methods to determine CIs exposure to floods (e.g . Lyu et al., 2018) and direct flood impacts (Winter et al., 2016; Kellermann et al., 2016). Despite indirect impacts and cascade effects are widely assumed as 35 more significant due to the interconnected nature of networks (Gil and Steinbach, 2008; Pant et al., 2016; Arrighi et al. 2017), few works are available which address indirect impacts and cascade effects in time and space (Pant et al. 2016, Arrighi et al., 2017). Among these works, indirect impacts and cascade effects are mostly addressed with complex conceptual frameworks (Fekete, 2019; Emanuelsson et al., 2014), simplified risk indexes(Lyu et al., 2018; Balijepalli and Oppong 2014; Singh et al. 2018) and/or very limited application to real-world case studies (Arrighi et al. 2019; Pant et al., 2016).



In order to address the above gap, this study aims at developing and applying a multi-infrastructure framework for the assessment of indirect flooding impacts. The framework is practically developed for addressing flooding impact to two linear CIs, namely WSS and the road system. Compared to point infrastructures, linear ones are more difficult to simulate because they have complex interconnections which induce a non-linear dynamics of impact propagation outside the directly flooded segments, thus they require ad hoc modelling. Three different models are respectively used for simulating inundation, water

pressure in WSS and traffic disruption due to flooding. Also, different spatial scales are considered to account for the dynamic nature of indirect impacts. The method is demonstrated for the metropolitan area of Florence (Italy). This study represents one of the first attempts to model flooding impact to CIs for real-world networks, considering mutual interconnections, and it is expected to be relevant to researchers, as well as practitioners.

### 1.1 Cascading effects in CIs and urban resilience

Modern cities are currently defined as "systems of systems", where the "systems" are Critical Infrastructure (CI) systems (Gardner, 2016). Infrastructure include all "the basic physical and organizational structures and facilities needed for the operation of a society or enterprise" (Hilly et al., 2018); CIs identify assets or systems which are so vital to a society that their incapacitation or destruction would debilitate security, economy, public health or safety, or any combination thereof (Serre and Heinzlef, 2018). Transportation systems, communication networks, sewage, water, and electric systems are all examples

of linear CIs.

The intrinsic nature of a "system" lies in the systemic interdependency, i.e. "a bidirectional relationship between two assets in which the operations of both assets affect each other" (Petit et al., 2015). Since an interdependency is effectively a combination of two dependencies, this constitutes a risk multiplier (Zio, 2016). Therefore, CIs cannot be considered independently, and silo-based analyses are completely inadequate to understand the behaviour of a given infrastructure operating in its

environment (Dueñas-Osorio et al., 2007; Rinaldi et al., 2001). These interconnections can lead to domino effects or cascading failures, i.e. the disruption or failure of a component in one infrastructure caused by the disruption or failure of a component in another infrastructure (Hilly et al., 2018).

The "system-of-systems" connotation comprehensively characterises CIs and the wider urban resilience. The resilience of cities depends on both the intra-system resilience (the resilience of the individual CIs) and on the inter-system resilience

(systemic resilience) (Kong and Simonovic, 2018). Hazard consequences may indeed extend well beyond the direct effect to the individual CI, and escalate disruption over physical and jurisdictional boundaries. Infrastructure can respond to an hazard in multiple ways by: *(i)* absorbing the impact and minimizing consequence; *(ii)* adjusting to non-optimal conditions and provide a lower performance (e.g. reduced service); *(iii)* failing in some or all its parts, without affecting other services; *(iv)* failing in some or all its parts and cascade the failure to other services (Kong and Simonovic, 2018).

A more thorough understanding of the complex interactions among CIs and relative consequences is therefore essential in preparing for, responding to, and recovering from disasters. The assessment of the consequences or impact depends on: *(i)* magnitude of service interruption (intensity, e.g. the number of impacted users); *(ii)* extent of service interruption (spatial); *(iii)* duration of service interruption (temporal) (Kong and Simonovic, 2018). The systemic interdependency has a strong temporal dimension (e.g. hazard time window, speed of recovery, repair time, disruption duration, emergency backup time),

as well as a spatial one (e.g. physical proximity, physical connection, hazard extent, spatial location of failures). In fact, indirect impacts usually dynamically propagate beyond the hazard domain, with an extension which is not proportional with the return period (Arosio et al., 2020). For other systems (e.g. finance, economy), Rinaldi et al. (2001) also considered logical interdependency, i.e. when the (intangible) connection is within variables and/or human behaviour. Appropriate metrics are essential to objectively measure both direct and indirect impacts; the impact due to a damaging event can be seen as the produce

between the duration of the event and the overall number of people affected (De Bruijn et al. 2019).





Advanced analysis that includes the systemic impact to interlinked CIs is needed to support national and local stakeholders in making better-informed and more holistic decisions, as underlined by The Sendai Framework for Disaster Risk Reduction (United Nations, 2015). Silo-infrastructure studies are limited in their scope since they ignore cascade effects and thus underestimate impact (De Bruijn et al. 2019).

Water Supply Systems (WSS) are essential for bringing freshwater; they are complex systems composed by a range of elements functional for collection, storage, transportation, treatment and distribution (e.g. pipes, water pumps, water treatment plants, reservoirs) (Bartram et al., 2009). The assessment of flood risk on a WSS requires a comprehensive approach including several scales of analysis (e.g. catchment area, riverbed, distribution network) and relative models (flood model, distribution network model). These models simulate pressure behavior in the nodes of the network if components supplied by electricity are affected

by flooding (Arrighi et al., 2017; Tarani et al., 2019);pressure fluctuations or low pressures may lead to contamination from leakage orifices and air vacuum valves (Ebacher et al., 2010; Ellison et al., 2003). Existing works implemented methods which integrate GIS analysis, inundation modelling, and hydraulic network modelling with Pressure-Driven Demand (PDD) (Cheung et al., 2005; Siew and Tanyimboh, 2012; Tarani et al., 2019). Two metrics measure flood impact to the WSS operativeness and integrity: *(i)* the number of inhabitants experiencing lack of service; *(ii)* the total length of potentially contaminated pipes

(Arrighi et al., 2017).

Among CIs, roads are also fundamental for everyday needs of mobility, delivery and accessibility; particularly, during emergencies they become critical for medical supplies, rescue and repairs, with some road links more crucial than others when looking at accessibility (Balijepalli and Oppong, 2014). Network analysis is a powerful tool to navigate citizens, civil protection operators and rescue teams to a chosen destination using a specific journey. Thanks to analysis within routing, it is

possible to decide if a route would be the shortest (length), the quickest (time) or safest (e.g. less flood-prone, more protected) according to specific needs. For example, an ambulance would rather use the quickest path, while citizens with normal cars could prefer the least vulnerable roads. Routing calculation can improve static vulnerability assessment and integrated analysis improves planning of accessibility to critical infrastructure (hospitals, shelters) during flooding (Arrighi et al., 2019; Pregnolato et al., 2017a). In the context of flooded roads, impact assessment criteria include: *(i)* number of flooded links; *(ii)* timing (flood

duration, operation time, traffic dynamics); and *(iii)* level of performance (e.g. speed reduction, road capacity reduction) (Balijepalli and Oppong, 2014).

These silo-based studies investigated road links' vulnerability and their failure impact on the overall network functioning (that is crucial for emergency planning), however they ignore the impact of disconnection and lack of accessibility to critical services which lead to cascading effects. In the last decade, a bulk of research has approached CIs interdependency and pioneered

methods to assess cascading effects. Review papers are available in literature for a more comprehensive overview (e.g. Ouyang, 2014). Modelling examples of flooding impact to interlinked CIs include: electricity, roads and civil protection infrastructure (Fekete et al., 2017); roads, IT, water and energy supply (Kong and Simonovic, 2018); energy, water and wastewater (Holden et al., 2013); electricity, airports, waste water plants and telecommunication (Pant et al., 2018); sewer and roads (Dong et al., 2019); water and energy supply (Byers et al., 2015). Other studies adopted a participatory approach and investigated

interdependencies between infrastructure/services in cities via workshops and/or interviews of residents and utility operators (e.g. De Bruijn et al., 2019). Despite such progress, how cascading consequences propagate, interact, trigger, and particularly what are the interdependencies at their spatial and temporal scales are still open research questions (Pescaroli et al., 2018). In fact, most of existing multi-infrastructure frameworks are limited in their scope since they remain very theoretical or conceptual (this includes network-theory-based studies, e.g. Li et al., 2018), infrastructure parameters are treated as random variables and

thus they lack of real-world application. Very few studies (Pant et al., 2018; Dong et al., 2019) developed a truly holistic application to analyse interdependency effects; however, indirect consequences are not investigated, especially regarding the WSS-roads interaction.





### 1.2 Motivation and aim

Infrastructures have currently to cope with the increase of population, aging of the assets, climate change and inherent
complexities (interdependency, technology) of modern cities (Holden et al., 2013). As a result, modellers and decision-makers
are looking for advanced methods to assess cascade effects for planning and risk management (Alexander, 2018; Pregnolato
and Dawson, 2018), to produce robust strategies for efficient performance in the future (any future). Existing studies offer
well-established methods to determine CIs exposure and direct flood impacts for flooding scenarios; however, a limited
number of studies is available about indirect impacts and cascade effects. Moreover, most of these works offer complex
conceptual frameworks, without application to real-world situations. In particular, existing models could be improved by
exploring the impact to CIs due to a real, weather-related hazard by integrating different models, e.g. a flood model and a GIS
system (Holden et al., 2013).

Flooding is a threat for any infrastructure system in urban environments. Floodwater can directly affect by physically damaging
assets and equipment (e.g. road pavement, power generators), and these damages can reflect into further disruption to other
system that rely on such damaged assets and equipment. For example, floodwater can shorts-out the energy supply (direct
impact), which leads to disruptions into the systems working with electricity: water supply (e.g. water pumps), transport (e.g.
railway, signalling); communication (e.g. routers), waste water (e.g. treatment plants), etc. Figure 1 shows the direct and
indirect impacts within the urban infrastructure systems due to a flood event. Hilly et al. (2018) explicitly highlighted that an
example of cascading effect from floods is when the WSS cannot be repaired because the roads that provide access to the
affected parts are flooded. This is the gap on which this paper focuses (Figure 1).

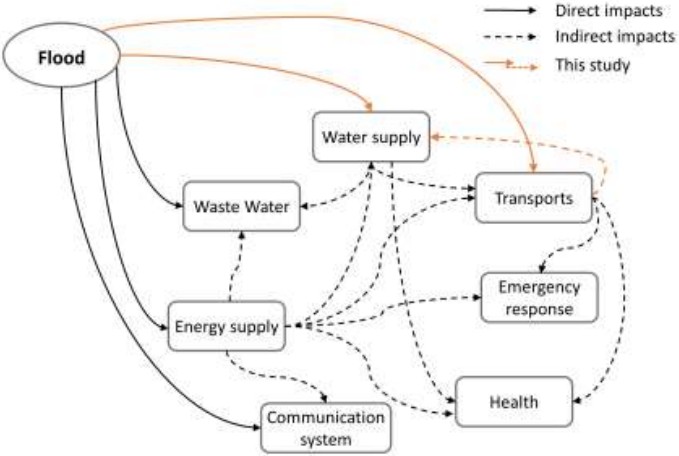

**Figure 1: Direct and indirect impacts on urban critical services and infrastructures. The orange lines identify the focus of this paper, i.e. the interdependency between WSS (Water Supply System) and the road system.**

This study aims at developing and applying a multi-infrastructure framework for the assessment of indirect flooding impacts.
This aim is achieved by: *(i)* proposing a risk-based approach which integrate multiple CIs, their interdependency, direct and
indirect impacts; *(ii)* modelling flooding indirect impact to two linear infrastructure systems (WSS and roads), after identifying
metrics which are compatible and representative for both networks, at both silo-based and interdependent level; *(iii)* modelling
the actual WSS and roads of the city of Florence for four flooding scenarios as a proof of concept, with a focus on the
consequences on WSS due to the lack of accessibility; *(iv)* drawing lessons and recommendation on the results, e.g. by
comparing methods and results from the silo-based and independent analysis.




## 2 Materials and method

Risk is traditionally described with four moduli: hazard, exposure, vulnerability and consequences (Grossi and Kunreuther, 2005). This study adopts this risk approach to develop a comprehensive methodological framework (Fig. 2) for computing the

risk to roads and WSS due to the indirect impact of flooding. Flood hazard includes the probability of a flood event of a certain intensity to happen; exposure represents the assets subject to flooding (road network and WSS); vulnerability is the extent of impact under certain conditions of exposure and hazard, e.g. the population not served by these assets. Direct impacts are impacts that may occur due to the physical impact of water, e.g. structural failure of bridges or instrumentation failure; direct impact assessment is out of the scope if this paper. Indirect impacts include service disruption, such as consequences of

travelling delays due to floodwater on the roads or pressure fluctuation due to malfunctioning of lifting stations. Pressure fluctuation and zero pressure in pipes lead to the entrance of undesired pathogens with consequent contamination especially in older networks. Indirect impacts are computed with ad-hoc models (Sec. 2.2) that concern the studied infrastructure systems (WSS and roads); however, the impact analysis could be tailored with different models for exploring impacts to other infrastructures (e.g. power supply and communication). The identified interdependency, which is a key element in cascade

effects, concerns the reduced performance of the flooded road system, which do not allow access to WSS main plants for repairs and replacement of the WSS elements damaged by floodwater. Considering the Annual Average Loss (AAL) as a reference risk metric, the risk is finally computed by estimating the impacted road length per year and the contaminated pipes length per year, as well as the delayed travellers and the not supplied Population Equivalent per year respectively.

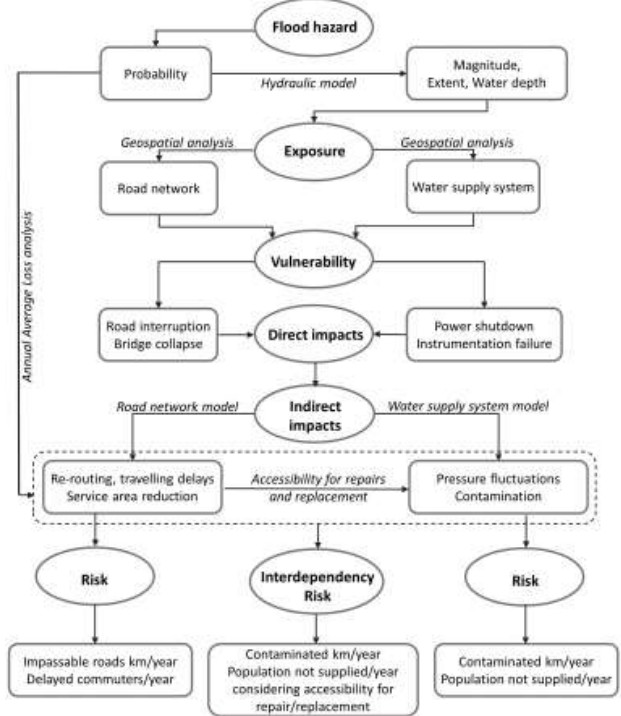

**Figure 2. Methodology flowchart.**

### 2.1 Flood hazard, exposure and vulnerability analysis

Flood hazard is represented by a map of flood parameters (intensity measures), usually water depths and flow velocity, for assigned probabilistic scenarios. Exposure is evaluated by intersecting hazard maps with GIS elements of the road and WSS networks. WSS nodes are assigned corresponding flood depths through raster zonal statistics. Active nodes, e.g. pumps or





lifting stations, exposed to the flood, are modified in the WSS model and marked as failed to simulate the indirect impacts to the network (see Sec. 2.2). Similarly, road network segments, except those described as bridges or embankments, are assigned the flood depths for each inundation scenario.

This study considers crucial to identify metrics which are compatible and representative for both road network and WSS (Table 1); this compatibility allows to compare results for the silo-based analysis and, more importantly, to develop the interdependent

analysis.

**Table 1. Hazard, exposure, vulnerability and impact metrics for the studied linear infrastructures: WSS (Water Supply System) and road network.**

|  | **WSS** | **Road network** |
|---|---|---|
| Hazard | Water depth (WD) (m) | |
| Exposure | WSS nodes | Road segments |
| Vulnerability | WD=0m: 100% operational WD>0.25 m: no operational | WD=0m: 100% operational 0m> WD > 0.3m: partially operational WD≥ 0.3m: no operational |
| Consequences (indirect impact) | Length of potentially contaminated pipework (km) | Length of impassable roads (km) |
| | Population Equivalent (PE) not served | Delayed commuters (PPH) |

Regarding the hazard, water depth (m) is commonly referred as intensity measure for flooding and adopted in this study. The

exposed elements include the elements of the WSS and road system susceptible to water depth, respectively WSS nodes and road segments (including embankments, elevated roads and underpasses). Vulnerability is defined according to the type of node and road, through the relationships shown in Table 1: *(i)* for WSS, a binary function determines a pumping station non-operational when this is touched by floodwater (Tarani et al., 2019); *(ii)* for the road network, a depth-disruption function computes the reduced velocity for vehicles travelling in flooded roads up to 0.3 m, i.e. the threshold of roadworthiness

(Pregnolato et al., 2017b).

## 2.2 Indirect impacts and cascading assessment

On the basis of literature (see Sec. 1), two impact metrics are selected for assessing indirect impacts: *(i)* the length of the disrupted network; and *(ii)* the population which experiences loss of service. The first metric can be easily converted into economic costs, when replacement/repair costs per unit length are available; the second metric is more suitable to describe

unmonetizable social costs.

For the WSS, the metrics are the length of potentially contaminated pipework and the Population Equivalent (PE) not served. PE is commonly defined as the ratio of the sum of the daily total demand to the individual demand of one person. In Italy the individual demand in cities for WSS design is prescribed by technical law and is about 200 l/day per inhabitant in average size cities. PE would coincide with the number of inhabitants for pure domestic water consumption. For the road network, these

two metrics are respectively represented by the length of impassable roads and the delayed commuters. When assessing disruption impacts to road and rail, People per Hour (PPH) delay is the metric used to account for both the time loss and the number of people affected (e.g. https://bit.ly/3kCj6tO): by instance, a PPH is the number of hours a public transport service is delayed multiplied by the number of users or passengers.

The WSS model is based on the freeware EPANET software, which calculates pressures at the nodes given a set of initial tank

levels, pump switching criteria, base nodal demands, and demand patterns. The standard EPANET simulations describe a



demand-driven approach, which stems from the direct goal of simulating correctly operated networks. When simulating the failure of the network (e.g. in the case of the inundation of the main lifting station), a Pressure Driven Demand (PDD) approach is more appropriate since the nodal demand are not attributed a priori; in fact, their value depends on the current local pressure. It is thus possible to analyse the temporal dynamics of the pressures in the network and consequently the population not served.

Further details on the PDD methodology can be found in Arrighi et al. (2017).

An integrated model that couples flood disruption and transport network is adopted to assess changes in time and distance in users' journey (Pregnolato et al., 2016; Pregnolato et al., 2017a). Floodwater reduces travelling speed of vehicles or stops traffic flows: roads are commonly considered closed when the flood depth reaches 30 cm (i.e. the depth at which a standard passenger's car is unable to operate), underpasses and elevated roads are considered as closed or completely functioning

respectively. The model evaluates the disruption to network links by comparing pre- and post-event travel times, thanks to a depth-disruption function (Pregnolato et al., 2017b) and a GIS–based accessibility model (Ford et al., 2015). For the flooded scenarios, the network properties of a link (i.e. travelling speed) are modified according to the functions, and traffic parameters recalculated for the perturbed state. Subsequently, journey travel time will increase in comparison with the baseline scenario, and eventually journeys are re-routed according to the quickest path (Dijkstra's algorithm - Dijkstra, 1959). A network Service

Area (SA) defines a region that encompasses all accessible streets within a specified time impedance (e.g. a 5-minute-impedance SA of a point includes all the streets that can be reached within five minutes from that point); they can also be defined as isochrones of equal travel time. Pre- and post-event SAs helps to evaluate accessibility and accessibility changes due to flooding.

This study investigates the additional secondary impacts due to the interdependence between road system and WSS by

assessing the lack of accessibility to critical elements of the WSS such as pump stations, preventing repairs and replacements. The delay or the lack of repair/replacement greatly undermines the post-event recovery time. The widely accepted definition of resilience is the ability of overcome an impactful event and return to normal condition through a quick recovery; the post-event recovery time is indeed a key metric of resilience. As shown in Figure 3, the pre-event condition (segment 1-2) is the business-as-usual performance of the system where preparedness actions might be adopted. When the natural hazard occurs

(2), the state of the system is altered by the impacts of the natural phenomenon, which lead to a reduction of the system performances (e.g. reduced road trafficability, reduced water pressure in pipes). The drop and recovery of the state may have different shapes according to the resilience of the system and the incorporation of the knowledge of cascade effects and interdependencies in preparedness and emergency plans. A low-resilience system, where preparedness and emergency plans are developed with a silo-based approach (c), is subjected to a sensibly higher reduction of system performances (2-3''). In

fact, unplanned emergency repairs or the activation of backup systems imply a slower recovery (3''-4''). This slow recovery could be due to the delays in assistance caused by the inaccessibility of WSS lifting stations due to flooded roads (silo-based), as well as a slow activation of emergency actions (low resilience).

Again, when emergency actions are promptly activated (high resilience) but there is a poor understanding of infrastructure interdependencies (silo-based) (b) there are delays due to unpreparedness (2-3').

If plans are developed with a systemic approach, cascading effects can be timely tackled (2-3) and the recovery phase shortens, reducing indirect impacts and associated costs. Recovery time relates to the performance loss of the system because post-event recovery operations and necessary resources increase over time, e.g. disinfection-flushing of WSS pipes for an increasingly larger portion of the network.

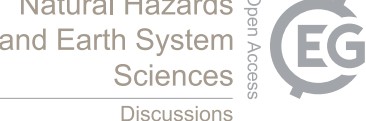



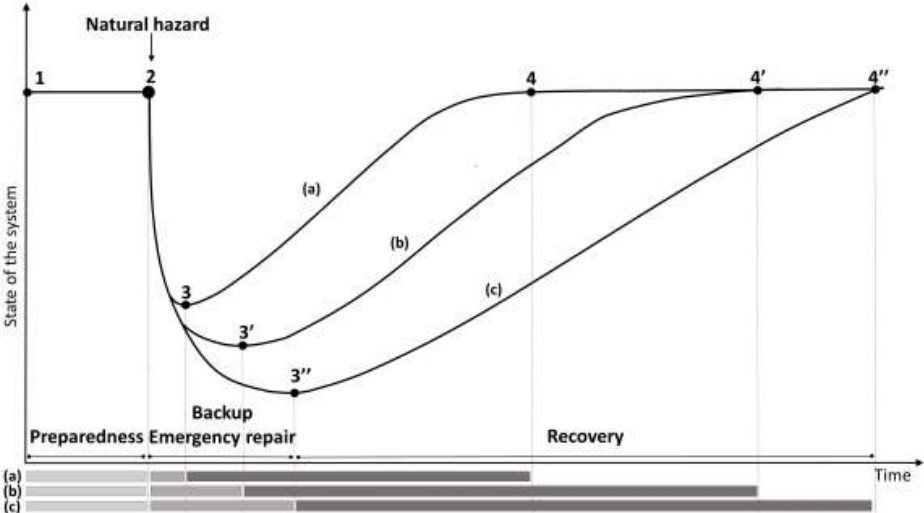

**Figure 3. Analysis of time windows pre-, during and post- flood event for high resilience with system-based management approach (a), high resilience with silo-based approach (b), low resilience silo-based approach (c). The delay or the lack of repair/replacement greatly undermines the post-event recovery (c).**

Timing is crucial for resilience to natural hazards; in a context of flood especially, adequate intervention can follow emergency plans which are activated by Early Warning Systems. The identification of the most vulnerable pumping stations and streets allows to plan accordingly in advance, by e.g. retrofitting some stations or developing emergency plans. The method developed by this study supports informed preparedness, which includes the identification of the pumping station(s) which should be reached in useful time for repair/replacement, reducing the breakdown time, increasing the speed of recovery and decreasing cascade effects.

## 2 Case study

The methodology is applied to the metropolitan area of Florence (Tuscany, Italy; 1 Mio inhabitants, of which about 37% corresponds to the city of Florence). The area has a long record of floods since the Middle Ages and is still prone to inundations (Arrighi et al., 2018). For frequent events, only few municipalities downstream of Florence are inundated whereas more severe scenarios (e.g. 200-year-event) affect the whole area, including the historic centre of Florence. From the hydrological point of view, this area is placed in the middle of the Arno catchment where the terrain morphology becomes flatter, the floodplain wider and the riverbed is more affected by anthropogenic changes, i.e. crossings, contractions, rectifications. The wider metropolitan area of Florence includes the three provinces (counties) Florence (FI), Prato (PO) and Pistoia (PT) (around 1000 $km^2$ of extent), and the province of Florence covers about 40% of the area. The municipality of Florence (city of Florence), the regional and Arno catchment Authorities are involved in the flood risk management of the area, which accounts for prevention, mitigation and preparedness measures. In this work, the Arno River Basin Authority provided hazard information (Autorità di Distretto Appennino Settentrionale, 2016), i.e. water depth maps for four return periods (Tr) (30, 100, 200, 500 years). The maps were developed by running a coupled 1D-quasi 2D hydraulic model in HEC-RAS. The river is simulated using the standard 1D-solution of the De Saint Venant equations and it is connected to the floodplain through lateral structures described by weir laws. The floodplain is modelled as a system of interconnected storage cells with mass conservation and stage-volume relationships. Terrain altimetry is described by a 1 m resolution, LiDAR-derived Digital Terrain Model (DTM) with 0.15 m vertical accuracy. More details can be found in Arrighi et al. (2013). The flood map for the worst-case scenario, i.e. 500-years recurrence interval, covers an area of 58 $km^2$ that includes 11 municipalities within the metropolitan area.




The municipal WSS features one main treatment facility with pumping station, 17 storage tanks, 619 km of pipework connected by 4863 main nodes to supply drinking water for domestic and industrial use (Figure 4. Presentation of the study area: (a) Florence metropolitan area and its commuter catchment (Regione Toscana, 2015); (b) municipal WSS nodes, main treatment

and lifting station; the road network system and Tuscany administrative boundaries (Regione Toscana cartographic portal). Abbreviations: MS - Massa Carrara, LU - Lucca, PT - Pistoia, PO - Prato, FI - Firenze, AR - Arezzo, PI - Pisa, SI - Siena, LI - Livorno, GR - Grosseto.b). The number of people relying on the municipal WSS are about 800,000 and could reside outside the municipal administrative boundaries. With non-domestic use the Population Equivalent of the system is about 874,000 (about half serving the non-permanent supply to Prato). In fact, the WSS provides freshwater to seven outer municipalities,

namely Bagno a Ripoli, Calenzano, Campi Bisenzio, Impruneta, Prato, Sesto Fiorentino and Scandicci (Fig. 5c), to increase the WSS redundancies and tackle potential drought situations. The connections to Calenzano, Campi Bisenzio, Sesto Fiorentino and Scandicci are stable and supply the whole demand of these municipalities, whereas the WSS connection to Prato activates only in case of insufficient availability of the city groundwater-based water supply. The two municipalities of Bagno a Ripoli and Impruneta are partly supplied by the Florence WSS, which integrates other local sources. The municipality

of Florence underpins a wide commuter catchment, i.e. areas from which the city attracts people for working purposes. The municipality of Florence has the highest number of residents that daily (internally) commute and the biggest commuter catchment in Tuscany (3490 km$^2$, 1,846,882 people); it includes 43 municipalities, 12 of which reside within the jurisdiction of other cities, i.e. Prato, Arezzo, Pistoia, Pisa, Livorno (Regione Toscana, 2015). The 79.5% of commuting journeys are within 30 minutes (the highest percentage in the region) and the most common mean of commuting is private cars (63%). The

road network features more than 13,000 km of roads and more than 670 km in the (simulated) hazard area.

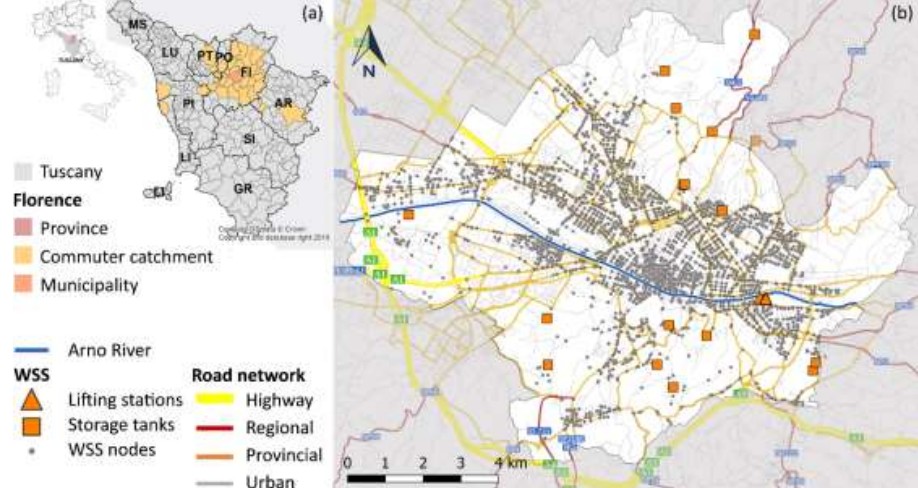

**Figure 4. Presentation of the study area: (a) Florence metropolitan area and its commuter catchment (Regione Toscana, 2015); (b) municipal WSS nodes, main treatment and lifting station; the road network system and Tuscany administrative boundaries (Regione Toscana cartographic portal). Abbreviations: MS - Massa Carrara, LU - Lucca, PT - Pistoia, PO - Prato, FI - Firenze, AR - Arezzo,**
**PI - Pisa, SI - Siena, LI - Livorno, GR - Grosseto.**

## 3 Results

### 3.1 Silo-based analysis

The case study analyses four recurrence interval scenarios. The central main pumping station, which draws water from the Arno river as main water resource, is not affected for low recurrence scenarios (i.e. Tr30 and Tr100). For Tr200 the failure of

the WSS lifting station causes the reduction of nodes pressures in the whole municipality, also far from the river and with effects on neighbouring municipalities. Initial storage tank levels are assigned with a warm-up simulation to reach a steady




state of the system; these initial levels allow maintaining a partial functionality of the system. After 60 minutes from the inundation of the lifting station, the 23% of the PE is not served and about one third of the pipes undergoes potential contamination. After 120 minutes, the nodes able to serve the nominal demand are limited to those supplied by storage

reservoirs in the periphery of the Florence municipality (i.e. three fifth of the nodes down, 47% of PE not served) (Fig. 5a). Three nodes linking to the municipalities of Sesto F.no, Prato and Scandicci fail and the node to Bagno a Ripoli has a residual partial functionality (Fig. 5a and b). If no remedial actions are taken, after six hours from the onset of the event, the disruption reaches 90% of the nodes and insufficient pressure affects 70% of the Florence PE (i.e. 268,100 people only in one municipality, three times the flooded population). All the links to outer municipalities are cut off with the highest impacts for

those municipalities which are not served by other sources and completely rely on the Florence WSS (Fig. 5c). Downstream neighbourhoods are served by local water tanks; also, they are favoured by lower terrain elevations, hence their nodes are affected later than those in the city center (Fig. 5a, green dots). 428 km of pipework in the network, i.e. the 68% of the whole WSS, is potentially contaminated due to null water pressure in this flood scenario.

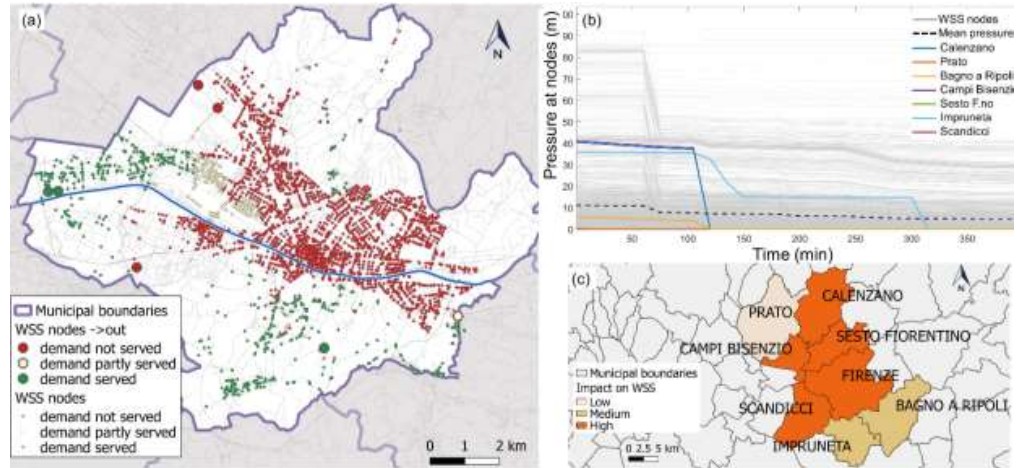

**Figure 5. 200 years recurrence interval flood impact on WSS. Pressure at nodes 120 minutes after the lifting station shutdown in Florence (a); temporal dynamics of pressure at nodes in the WSS system with detail of node links to outer municipalities (b); severity of impact in all the municipalities (administrative boundaries source: Regione Toscana cartographic portal).**

Table 2 shows the impacts for the analysed scenarios in the eight interconnected municipalities, six hours after the shutdown.

The affected population equivalent (PE) is larger than actual resident population in some municipalities, since there is also an industrial water demand, especially in Scandicci, Sesto F.no and Campi Bisenzio. The municipality of Prato, which is not connected to the network in normal situations, is considered not affected.

**Table 2. Flooding impact on the WSS network six hours after the shutdown, expressed as affected population equivalent (PE).**

| | | Tr 30 | Tr 100 | Tr 200 | Tr 500 |
|---|---|---|---|---|---|
| **Municipality** | **Tot. resident population** | **Affected PE** | **Affected PE** | **Affected PE** | **Affected PE** |
| Florence | 378104 | 0 | 0 | 264673 | 264673 |
| Bagno a Ripoli | 25481 | 0 | 0 | 18976 | 18976 |
| Calenzano | 17940 | 0 | 0 | 17787 | 17787 |
| Campi Bisenzio | 47141 | 0 | 0 | 50078 | 50078 |
| Impruneta | 14635 | 0 | 0 | 4700 | 4700 |
| Prato | 194590 | 0 | 0 | 0 | 0 |
| Scandicci | 50604 | 0 | 0 | 52293 | 52293 |
| Sesto F.no | 49331 | 0 | 0 | 51736 | 51736 |


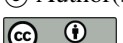



Similarly, for the WSS analysis, the users of the road network are impacted by flooding within and outside the inundated area. For example, for a 1-in-200-year event, around the 78% of roads in the hazard area is flooded, including 13 km of highways and 45 km of major roads (Figure 6 and Table 3).

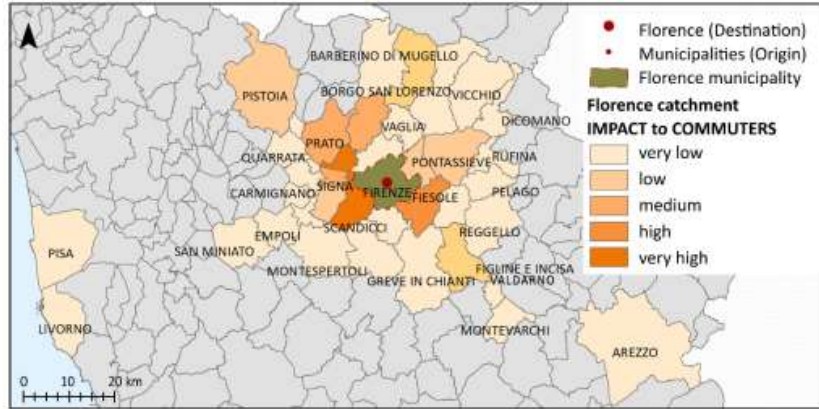

**Figure 6. The road network is impacted by flooding within and beyond the inundated area for a 1-in-200-ys event: sensibly, the municipalities of these travelers are those in the hazard area (Signa, Campi Bisenzio and Poggio a Caiano). However, impact areas include also municipalities which are not contiguous to the hazard domain. Considering People per Hour delay (PPH), the legend read as: very low: up to 100k PPH; low: 101k-250k PPH; medium: 251k-1000k PPH; high: 1001k-1500k PPH; very high: >1501k PPH. Administrative boundaries source: Regione Toscana cartographic portal**

More than 10% of commuters (> 195,000 people) are affected by an increase of their travelling time by 50%; reasonably, the municipalities of these travellers are those in the hazard area (Signa, Campi Bisenzio and Poggio a Caiano). The travelling time of the commuters of 29 municipalities (>63% of all commuters, >1 million people) are increased by >10%. However, impact areas include also municipalities which are not contiguous to the hazard domain, in the North (e.g. Calenzano, >132,500 people, ~30% time increase) and municipalities in the South East (e.g. Bagno Ripoli, >167,700, ~40% time increase).


**Table 3. Flooding impact on the road network, expressed as Population Per Hour delay (PPH). The ten most-affected municipalities are shown; all results are shown in the Appendix.**

|  |  | Tr 30 | Tr 100 | Tr 200 | Tr 500 |
|---|---|---|---|---|---|
| Municipality | Tot. resident population | PPH | PPH | PPH | PPH |
| Campi Bisenzio | 159589 | 0 | 10891 | 1944634912 | 7882483382 |
| Scandicci | 364049 | 2558 | 2558 | 31313 | 6650759773 |
| Sesto Fiorentino | 361921 | 0 | 0 | 974 | 6611777254 |
| Prato | 207337 | 0 | 6667 | 13183 | 3787762078 |
| Bagno A Ripoli | 167747 | 0 | 0 | 14576 | 3064512368 |
| Calenzano | 132519 | 0 | 4261 | 8426 | 2420940029 |
| Fiesole | 48278 | 0 | 0 | 2417 | 881972060 |
| Impruneta | 46847 | 77 | 77 | 1173 | 855836978 |
| Lastra A Signa | 40527 | 1175 | 2584 | 4063 | 740384966 |
| Pontassieve | 37265 | 0 | 0 | 2020 | 680779988 |

The Annual Average Loss (AAL) is finally computed by estimating the impacted road length per year and the contaminated
pipes length per year, as well as the delayed travellers and the not supplied equivalent population per year respectively (Table 4). 52.1 km of roads and 3.2 km of pipes per year are affected in the study area. About 260,000 commuters and 3500 PE are





yearly affected on average. The comparison between the impacts of the two networks clearly shows that, while roads are increasingly affected with increasing return periods, the WSS has the same impact for the two more severe events. In fact, there is a threshold behaviour triggered by the inundation of the lifting station, which is almost independent from inundation

extent and propagates in the system.

**Table 4. Flood impacts for single scenarios and risk in terms of Annual Average Loss (AAL) to road network and WSS in the study area considering a silo-based approach. (Abbreviations: PPH - Population Per Hour delay; PE – Population Equivalent).**

| Flood Scenario | Affected road (km) | Affected commuters | PPH (x10³) | Affected pipes (km) | Affected PE |
|---|---|---|---|---|---|
| Tr 30 | 84.8 | 481863 | 0.55 | 0 | 0 |
| Tr 100 | 373584 | 773971 | 39.4 | 0 | 0 |
| Tr 200 | 573642 | 1357180 | 1944737.5 | 428 | 460243 |
| Tr 500 | 713439 | 1357180 | 38707129.4 | 428 | 460243 |
| | | | | | |
| **Risk** | km/year | commuters/year | PPHx10³/year | km/year | PE/year |
| AAL | 52.1 | 259665.6 | 143257.2 | 3.2 | 3451.8 |

**3.2 Systemic analysis**

The system analysis includes the modelling and assessment of indirect flooding impacts cascading through the WSS and road

system of the city of Florence. In specific, this study evaluates the consequences on WSS due to the lack of accessibility, which prevents timely repairs and replacement at the WSS lifting stations.

The WSS lifting stations should ideally accessed for repairs before 60 minutes from the onset of the event because: (i) about 33% of the pipes undergoes contamination (Sec. 3.1); (ii) ca 90% of the street becomes impassable (by reaching the 30 cm threshold of roadworthiness). At latest, WSS lifting station needs to be repaired in 120 minutes, i.e. the back-up time provided

by storage reservoirs; sites in need of repairs would be reached by special vehicles, e.g. SUVs of the Civil Protection. The interdependency induces systemic consequence which is higher than consequences of the single WSS or road system alone. In this case study, the systemic consequence consists in the increase of the recovery time, due to extended outage (due to the lack of repairs), which implies a lower resilience of the city (see Fig. 3). If preparedness and emergency plans are developed with a systemic risk knowledge, i.e. with a full understanding of cascade effects, interdependencies among networks and repairs

before 60 minutes from the onset of the event, the AAL of the WSS reduces to 982 PE/year (-71.5%) and 1.5 km/year (-53.1%) in terms of PE and pipe length respectively. With remedial actions before 120 minutes, the AAL reduces to 2007 PE/year (-41.8%), and 2.7 km/year (-15.6%.) (Table 5).

**Table 5. Risk to WSS in terms in terms of Annual Average Loss (AAL) for different intervention times (Abbreviations: PE –**
**Population Equivalent).**

| Risk to WSS (AAL) | Intervention time (minutes) | | |
|---|---|---|---|
| | 60 | 120 | 360 |
| | | | |
| km/year | 1.48 | 2.69 | 3.21 |
| PE/year | 982.29 | 2007.28 | 3451.82 |

Service Areas (SAs) are applied to understand which portions of the city are accessible within a given time, i.e. the impedance time. The impedance time for this case study is 8 minutes, as prescribed by the Italian emergency measures; this impedance time would allow for timely repairs. The SAs were calculate for the four hazard scenarios, considering 8 minutes of travelling

(Figure 7) and the max depth of water along the whole network. In baseline condition (no disruption), both lifting stations A


and B are easily accessible from most of the Florence municipality and beyond; the SA measures around 120 km$^2$. For a 1-in-30-year event, the SA reduces to ca. 113 km$^2$ and both stations are still well-connected with the city. For a 1-in-200-year event, the accessibility level drops and the SA shrinks to less than 5 km$^2$: the lifting station B is no longer accessible, whereas station A can be reached from very few areas of Florence. For the most extreme event, i.e. a 1-in-500-year event, the SA is almost null and none of the stations are accessible.


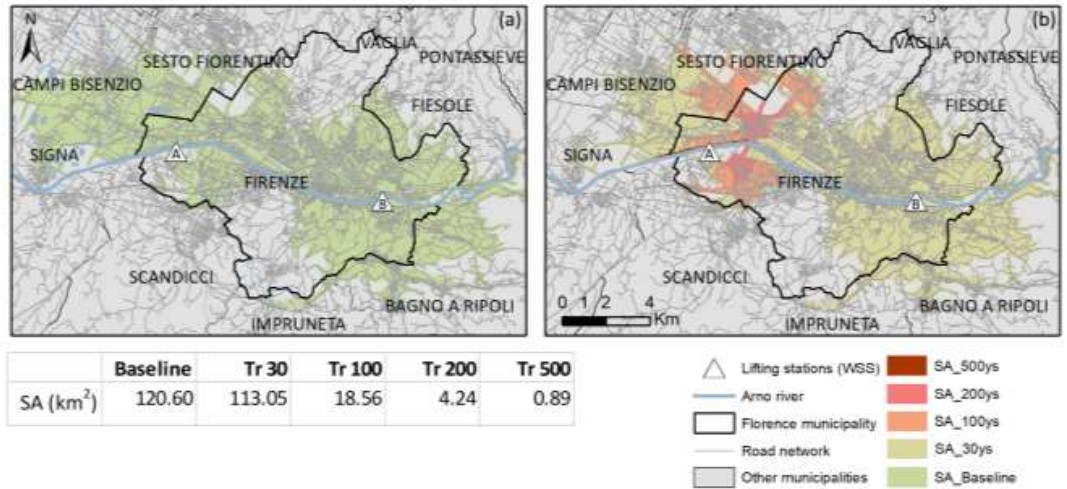

| | Baseline | Tr 30 | Tr 100 | Tr 200 | Tr 500 |
|---|---|---|---|---|---|
| SA (km$^2$) | 120.60 | 113.05 | 18.56 | 4.24 | 0.89 |

**Figure 7. Accessibility analysis for the lifting stations of the WSS system using Serviced Areas (SAs). The figure shows spatial visualization for: (a) the baseline (no flooding) and (b) the four considered hazard scenarios. The A and B triangles are the lifting stations of the WSS. Administrative boundaries and road network source: Regione Toscana cartographic portal**

Results show that for extreme scenarios (Tr200, Tr500), road conditions do not allow to reach both lifting stations on time, unless emergency vehicles are sent before critical depths are reached. Such operation would happen if the city is prepared to act upon flooding events and actions are pre-planned. Although a lifting station is usually equipped with a backup engine-generator to prevent from power outages, this cannot work in submerged conditions; moreover, the flood duration might be be longer the duration of fuel availability for the engine-generator. The first recommendation is then to develop *ad-hoc* emergency

plans by identifying potential critical hotspots (e.g. WSS lifting stations), and to analyse the accessibility to these sites. Secondly, to equip the WSS implants with vehicles for repairs that are better resistant than normal cars to water, e.g. 4WD or SUVs (like Civil Protection's ones), would allow to reach sites normally cut off by floods. An alternative option would be to equip the implants with a water-resistant back-up system, which increases the time window for repairs. The third recommendation is to enhance the system redundancy for those municipalities totally reliant on a single main system, with e.g.

emergency water storage tanks.

## 4 Discussion and future research

The WSS and road transport system are two linear infrastructure systems, so they are comparable in terms of flood hazard and impact metrics. For example, the indirect impact respectively included pipework to be decontaminated and length of network affected (i.e. closed or reduced speed segments for roads), as well as the population disrupted by the loss of each service.

However, the system's exposure to flood is measured differently, since the WSS system considers the number of nodes, while the road network features the road network length. Also, service disruptions can lead to different types of potential consequences: the population not served by WSS may experience health issues due to the lack of a basic resource, especially for more vulnerable categories; in the case of road network, delayed commuters may experience difficulties in reaching their workplace, which can be converted to monetary loss (e.g. using the Value of Time).





Moreover, the two infrastructure types have peculiarities in relation to the temporal and spatial scale. WSS disruption starts as soon as the lifting station (next to the river) is flooded (first significant pressure drop in the network in 60 minutes), while the road network is progressively flooded until reaching the maximum depths (3 to 6 hours depending on location), alongside incremental disruption. For the WSS, the indirect impacts reached eight municipalities and impacted all the users (460243 PE), excluding Prato municipality who does not rely on the nodes of the Florence municipality. For the road network, the indirect

impacts affected all the commuters travelling to work in the Florence municipality (1,846,882 people), covering also areas which are not contiguous to Florence (Pisa, Livorno, Arezzo). Commuters settled in the eight municipalities affected by the WSS failures are experiencing both disruptions; hence, if they decided to not travel to work upon an issued flood alert, they would experience lack of water in their homes – due to the same flood event.

This case study highlighted the importance of identifying spatial and temporal thresholds to cascading effects; eventual failures

of the WSS system should be addressed in a time window which includes the time to issue the alert, the time to reach the sites and the time to repair the equipment. For example, for the Florence case study, it would be ideal to repair the lifting station before 60 minutes the beginning of the event; in fact, after 60 minutes there is the first WSS pressure drop and the onset of the potential pipe contamination. An upper-end threshold would be to repair the WSS systems in 120 minutes, to avoid reaching the 70% of node pressure dropping; however, potential contamination would still affect part of the network. Timely repairs to

the WSS, assisted by back-up system, makes the difference between a high- and a low-resilient urban environment (Figure 3). This work focused on risk analysis, therefore it adopted the worst-case-case hazard map per each scenario, by considering the maximum depth for each point. Static hazard maps (with maximum depth) are also currently typical for urban planning. However, this approach is not fully satisfying when considering the complex dynamics of water runoff, water supply and traffic flow. A more advanced hydraulic modelling is suggested to produce dynamic hazard maps (with hazard timesteps), and

assist with a detailed modelling of infrastructure dynamics. In particular, this advanced modelling is seen necessary for identifying actual emergency operations and plans, looking at fitting the time windows of the disruption. For example, the 'dynamics' of flooding and accessibility are likely to significantly differ for Tr200 and Tr500: Tr500 magnitude is higher, and floodwater would flow away from roads much slower, increasing the recovery time.

Additional work could estimate the risk for a wider range of probabilistic scenarios and investigate how the two infrastructures

recover after the events. Further studies could also include the physical impact to road elements (such as bridges), mitigation measure for the WSS and the interdependencies with a third infrastructural system, e.g. power supply. Future studies could particularly focus on the temporal element and scale of the risk analysis, to estimate e.g. operations time. Finally, pressure changes in the WSS network can cause ruptures of aqueduct pipes with consequent water flows in the roads; this cascading effect could be a further topic of future research.

**5 Conclusions**

This work presented the risk analysis of two linear infrastructure systems, namely the water distribution system (WSS) and the road network system. The study presents a real-world application to the city of Florence (Italy). The evaluation of indirect flood impacts on the two networks was carried out for four probabilistic scenarios (30, 100, 200, 500 years), obtained by a coupled 1D-quasi 2D hydraulic model. A Pressure-Driven Demand network model and a flood-transport network model were

used for assessing the impacts on the WSS and the road network respectively. The analysis focused firstly on the identification of common impact metrics for both systems for hazard, exposure, vulnerability and (indirect) consequences. Secondly, the study adopted a silo-based approach to assess impact and risk to the WSS and road system, as separated entities. Results showed that the impact of flooding to the two systems differs in both spatial (up to 5 affected municipalities per WSS, 37 for the road system) and temporal scale (60 minutes before first pressure drop, 30 minutes to reach critical depths on roads).

Thirdly, a systemic approach was used to include interdependency and cascading effects, thus assess impact and risk at system-



level. When interdependencies are accounted for, results showed that the risk to the WSS in terms of Population Equivalent (PE/year) can be reduced by 71.5% and 41.8%, if timely repairs to the WSS stations are accomplished by 60 and 120 minutes respectively; the risk to WSS in terms of pipes (Km/year) reduced by 53.1% and 15.6%. The study highlighted that systemic risk-informed planning can support timely interventions and enhance infrastructure resilience; however, it is recommended to
conduct further studies which focuses on the complex dynamics of water runoff, water supply and traffic flows to support practical action planning.

**Data Availability Statement**

All relevant and public available data will be shared via the DataBris repository of the University of Bristol if the paper is accepted for publication; data sources are clearly specified throughout the paper.

**Author Contribution**

CA conceived the research work and developed the flood hazard, the vulnerability modelling of WSS, the AAL calculation for silo-based and systemic analyses. MP undertookthe road network modelling and the service area analysis. CA and MP wrote the manuscript, produced the maps and schemes. FC supervised and reviewed the work.

**Competing Interests**

The authors declare that they have no conflict of interest.

**Acknowledgements**

Maria Pregnolato was supported by the Engineering and Physical Sciences Research Council (ESPRC) LWEC (Living With Environmental Change) Fellowship (EP/R00742X/2). Authors acknowledge Publiacqua SpA for providing the sample network data and for the advice given as stakeholder. Source of GIS road and administrative data: Regione Toscana cartographic portal
(https://www502.regione.toscana.it/geoscopio/cartoteca.html)

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
