# Peer review of "Indirect flood impacts and cascade risk across interdependent linear infrastructures"

_Natural Hazards and Earth System Sciences, 2020_

## Referee Comment (RC1) · Anonymous Referee #1 · 15 Dec 2020

Overall a very relevant empirical paper that contributes to an area which is still lacking such studies.

On language I am not qualified to judge, but you may run a grammar check on some sections where "a" or "the" seem to be missing such as in line 25.

Abstract first sentence: check for usage of hazard instead of threat. I suggest to modify it to "one of the..." which seems more realistic given the wide spectrum of what could be regarded a natural threat/hazard. Earthquakes are more frequent, for example, but you probably mean a certain combination with magnitude.

Line 9: add "for example", since indirect effects occur not only through infrastructure.

Line 42-44. Linear structures can be come complex, but must not. Within complexity

theory, linear systems often are not understood as complex. Also I am not sure why point infrastructure should not become complex regarding interactions. Considering Rinaldi et al 2001 on interdependency types, I would assume that for example, logical interactions between point objects can become quite complex, i.e. non-linear. I therefore suggest to formulate it a bit more cautiously. Also, why must it be ad-hoc modeling and not before and event? "Three models are ..." you may add "in this study"

Line 47: I think this claim is not backed by a thorough literature analysis, so I suggest to omit it or make it more precise; related to the specific location. Some of the authors you have mentioned above, for example, have done such assessments, too. But many others, too. Line 48 provide some examples of fields where it could be applied.

Line 50: "Systems" are not just "CI systems"; correct the wording. Line 52: Some sources have rightly criticized that CI are more than just physical or organizational; they include staff, humans as user, environment, non-structural aspects such as regulations etc., too.

Line 70. A "more thorough understanding" should also go beyond traditional magnitude/probability formula, some argue and should include impacts in terms of different types of impact spheres (human, environment etc.) but also include questions of which quality, quantity and volume of values are affected and what types of risk management or protection goals exist to help prioritize such criticalities.

135: check the term "shorts-out"

166 Add a source to AAL. What about the maintenance and repair cars and teams that are mentioned above as a main motivation?

202: instead of the tiny URL, provide a proper source description. What type of railway is this etc.

302: sources for the quantitative measures? or is this computed by you? until 320: it seems you have computed those values; what data did you use to achieve it (i.e. road

network data sources, types etc.?)

Table2 very nice and relevant results. Could you provide estimates of possible error margins? A sentence or two would suffice, maybe in the discussion. Just since this looks to exact.

356 check grammar

359 do those SUVs permit higher wading depths, such as trucks (0,5m)?

399 This must not be so, road nodes and section, crossing could also be of interest. Maybe just add "within this study"

403 add a source for Value of Time, or VOLL

428-433 Interesting areas for future work. Might be illustrative for readers to add a few sources as examples who covers bridges, power supply, operation times etc.

Literature Some sources cited in the text are missing in the reference list.

---

## Referee Comment (RC2) · Anonymous Referee #2 · 13 Mar 2021

In this paper, the authors developed a risk analysis of the water distribution system (WSS) and the road network system under flooding events. The case study is the metropolitan area of Florence, which is in a flood-prone area. The paper aims to study the interdependence between the WSS and the road system by evaluating the accessibility to critical components of the WSS. Network models and topological metrics (e.g. the length of the disrupted edges, network service areas) are used to measure the vulnerability of the systems.

Overall, I found this paper interesting and relevant to the field of infrastructure resilience. In particular, the paper tries to analyse interconnections between two infrastructure systems by looking at a real-world case study. Anyway, the paper still needs

some work to be ready for publication. Therefore, I hope my comments will help in the revision process.

- The syntax of the whole text should be revised. In particular, I found sections 2 and 3.1 difficult to read. Moreover, I found many typo errors in the text (for example, line 234: (2-3"), line 388: "be be").

- In line 37, you presented previously published works and you wrote "Among these works, indirect impacts and cascade effects are mostly addressed with complex conceptual frameworks (Fekete, 2019; Emanuelsson et al., 2014), ..." How do you define a "complex conceptual framework"? Please, be more precise when reporting other works.

- Line 50: "Modern cities are currently defined as "systems of systems", where the "systems" are Critical Infrastructure (CI) systems (Gardner, 2016)." I do not agree with this definition. For me, a city is made also of people, cultures, the environment, the ecosystem, etc. Falco published a paper about this (Falco (2015) "City Resilience through Data Analytics: A Human-Centric Approach).

- In the text, you wrote often about the "impedance time". For example, in line 372: "Service Areas (SAs) are applied to understand which portions of the city are accessible within a given time, i.e. the impedance time." I have never heard about it. I checked on a vocabulary and it says it is related to electronic measures. Therefore, I am not sure if it is the most correct terminology.

- I found often "silo-based" in the text, but there is not a clear definition of it. I think it is important to add a definition because "silo-based" is a relevant concept for your work.

- Line 58: "Therefore, CIs cannot be considered independently, and silo-based analyses are completely inadequate to understand the behaviour of a given infrastructure operating in its environment (Dueñas-Osorio et al., 2007; Rinaldi et al., 2001)." This statement sounds a bit strong. I checked the two papers. Duenas-Osorio et al. (2007)

wrote in their abstract that "Effective mitigation actions could take advantage of the same network interconnectedness that facilitates cascading failures", while Rinaldi et al. (2001) wrote "When examining the more general case of multiple infrastructures connected as a "system of systems," we must consider interdependencies." You wrote in line 83: "Silo-infrastructure studies are limited in their scope since they ignore cascade effects and thus underestimate impact (De Bruijn et al. 2019)." I think that analyses of a single network system can advance our understanding of specific systems or they can help to find metrics to use for other analyses. In section 3.1, you also analysed the WSS and the road network separately. Based on those results you could measure the Annual Average Loss (AAL) on page 12. Overall, I think that the introduction should be revised from this perspective.

- In this paper, you used network models embedded in space for your analyses. Anyway, the paper did not report enough literature about this topic. Moreover, other papers studied the impact of floodings on road networks. For example, Casali and Heinimann (2019) "A topological characterization of flooding impacts on the Zurich road network. PLoS ONE 14(7)"; Kermanshah and Derrible (2017) "Robustness of road systems to extreme flooding: using elements of GIS, travel demand, and network science." Natural Hazards, 86.

- Line 120: "Very few studies (Pant et al., 2018; Dong et al., 2019) developed a truly holistic application to analyse interdependency effects; however, indirect consequences are not investigated, especially regarding the WSS-roads interaction." I found this sentence too strong. I would rephrase it because there are many published works that analyzed cascading effects on networks. Moreover, you reported the work of Dong et al. (2019), who developed percolation analyses on the road networks of different cities, not of interconnected networks. Therefore, why is a percolation analysis a truly holistic application for interdependency? I think that even the study of a single network system can represent a holistic approach since it looks at the network system as a whole. Casali and Heinimann developed a thesis from this concept (Casali

(2020), "Topological Assessment of Changes in Road Network Systems in Time, under Discrete Flooding Events, and under Classes of Unexpected Disruptions").

- Why do you use to measure indirect impacts: (i) the length of the disrupted network; and (ii) the population which experiences loss of service? Maybe in section 2.2 you can add more text about the motivations. Moreover, why is the total length of edges a better metric to analyse network vulnerability than other metrics (for example, the number of disrupted edges)?

- In the methodology section, I found that not all the information is reported fully. I do not find a definition for the Annual Average Loss (AAL) and details about how you calculated the PPH. Moreover, you introduced the Pressure Driven Demand (PDD), and it can be useful if you will add more information about it.

- In the methodology section, I did not find precise information about how you modelled the networks. For example, what are exactly a node and an edge in the WSS and the road networks? Which software did you use to model them? I understood that the road network extended to a larger area than the WSS network, is that correct? Did the road network add some weights to the edges? For example, in line 217, you wrote that "for the flooded scenarios, the network properties of a link (i.e. travelling speed) are modified according to the functions, and traffic parameters recalculated for the perturbed state." This means that you used the travelling speed in the analyses of the road network. Therefore, how did you calculate the travelling speed?

- You used the SA (network service area) to look at accessible areas. Did you consider also directions of roads when you analyse the shortest paths?

- Line 226: " The widely accepted definition of resilience is the ability to overcome an impactful event and return to normal condition through a quick recovery;" There are many authors that defined resilience in recent years. You can add a reference to a published work on infrastructure resilience.

- Figure 4: you can improve the resolution of the figure. Moreover, I cannot see the edges of the urban network in figure a.

- Section 3.1 "Silo base analyses": you can add a topic sentence to introduce the "silo-based analyses".

- Figure 5: I cannot read legend of figure b. The description of figure c is missing. You can improve the resolution of this figure. Moreover, what does it mean "low", "medium" and "high" in legend of figure c?

- Caption Table 2: I would repeat here that the WSS is not affected for 30 and 100 years events.

- Figure 6: it is not clear how did you choose the interval limits for this figure because it is not reported in the method section.

- Table 3: why there is not Florence in this table? Then, I would order the municipalities as in table 2.

- Lifting stations are important for the analyses of this paper. You can describe more the geography of lifting stations, for example, how many and where are lifting stations? Therefore, I ask the authors to provide more description of the topology of the WSS in the result section.

- In Line 389 you wrote, "The first recommendation is then to develop ad-hoc emergency plans by identifying potential critical hotspots (e.g. WSS lifting stations), and to analyse the accessibility to these sites." Therefore, you concluded that identifying critical components in a single network can help to improve emergency plans. Is my interpretation correct?

- Line 393: "The third recommendation is to enhance the system redundancy for those municipalities totally reliant on a single main system, with e.g. emergency water storage tanks." I agree but by adding new infrastructures maybe we add new externalities to the urban area? What are the immaterial costs of building new water storage tanks

in Florence? This paper might be interesting Chelleri and Baravikovab (2021) "Understandings of urban resilience meanings and principles across Europe".

- Conclusion part, line 444: "Results showed that the impact of flooding to the two systems differs in both spatial (up to 5 affected municipalities per WSS, 37 for the road system) and temporal scale (60 minutes before first pressure drop, 30 minutes to reach critical depths on roads)." I cannot find in the result part where you reported or discussed the 30 minutes to reach the critical depths on roads. Please, write this result in the main result section.

---

## Author Comment (AC1) · 12 Apr 2021

Overall a very relevant empirical paper that contributes to an area which is still lacking such studies.

Reply. Thank you for your comments. We appreciate that the reviewer recognized that the topic of cascade effects of natural hazards and associated indirect impacts is key for the flood risk community. Understanding the magnitude of potential losses of interconnected system can greatly improve preparedness and resilience of communities, thus more applied research should be carried out in this area.

On language I am not qualified to judge, but you may run a grammar check on some

sections where "a" or "the" seem to be missing such as in line 25.

Reply. A grammar check will be performed on the revised manuscript to correct typos.

Abstract first sentence: check for usage of hazard instead of threat. I suggest to modify it to "one of the..." which seems more realistic given the wide spectrum of what could be regarded a natural threat/hazard. Earthquakes are more frequent, for example, but you probably mean a certain combination with magnitude.

Reply. We are going to modify the sentence in the abstract to better recognize floods among other natural hazards.

Line 42-44. Linear structures can become complex, but must not. Within complexity theory, linear systems often are not understood as complex. Also I am not sure why point infrastructure should not become complex regarding interactions. Considering Rinaldi et al 2001 on interdependency types, I would assume that for example, logical interactions between point objects can become quite complex, i.e. non-linear. I therefore suggest to formulate it a bit more cautiously. Also, why must it be ad-hoc modeling and not before and event? "Three models are ..." you may add "in this study"

Reply. We agree that also point critical infrastructures are crucial and might interact with other points. A hospital can be seen as a point critical infrastructure and it is true that its manifold functions foresee the interaction with other points. What Authors meant here was that linear critical infrastructures (e.g. roads, piping systems, railways etc.) are complex networks of lines (arcs) and points (nodes); this networked nature makes them more susceptible to propagate impacts given both the physical and functional inter-dependency. To make this point clearer the paragraph will be rephrased and also the need for this study to use three models will be explicitly mentioned. This study shows an "ad-hoc" modelling to develop and apply a method; however, practical tools could be further develop from it, e.g. including modelling for pre-event assessment.

Line 47: I think this claim is not backed by a thorough literature analysis, so I suggest

to omit it or make it more precise; related to the specific location. Some of the authors you have mentioned above, for example, have done such assessments, too. But many others, too. Line 48 provide some examples of fields where it could be applied.

Reply. Authors carried out an extensive literature review on the topic and could not find any other paper which presented an applied work where indirect impacts and cascade effects are analyzed based on the dependence of critical infrastructure. Studies related to flood impacts on a single network or to conceptual interdependent networks were mentioned in the literature review (Section 1); this Section will be revised and updated with new works (if any).

Line 50: "Systems" are not just "CI systems"; correct the wording.

Reply. Authors adopted the definition of "systems" by Gardner (2016), which does strictly refer to an engineering and urban perspective. We are aware that other "systems" do exist (e.g. socio-economic, biological, etc.), but these are out of the scope. This aspect will be clarified in the manuscript.

Line 52: Some sources have rightly criticized that CI are more than just physical or organizational; they include staff, humans as user, environment, non-structural aspects such as regulations etc., too.

Reply. Authors absolutely agree, the perspective here described is more an engineering perspective related to 'hard infrastructures', but it is worth mentioning what you suggested. We will broaden the description in the text.

Line 70. A "more thorough understanding" should also go beyond traditional magnitude/probability formula, some argue and should include impacts in terms of different types of impact spheres (human, environment etc.) but also include questions of which quality, quantity and volume of values are affected and what types of risk management or protection goals exist to help prioritize such criticalities.

Reply. Authors recognize that a more thorough understanding may require a paradigm

shift going beyond the traditional approach. However, a disruptive change in evaluating impacts implies a wide agreement in the research community. Authors believe in the need of including robustly the societal sphere in these analyses but as engineers we need to involve other expertise in a multi-disciplinary work, often hard to achieve. We fully understand that considering the human sphere is key in understanding indirect (often intangible) impacts and this paper represents an attempt towards that direction. In fact, we presented different impact metrics where the people are central by evaluating the population not served and delayed commuters.

135: check the term "shorts-out"

Reply. The term will be substituted by "impair".

166 Add a source to AAL. What about the maintenance and repair cars and teams that are mentioned above as a main motivation?

Reply. AAL is the frequency-weighted amount of loss that can occur on average in any year, we will add a reference to a widely used manual by US-ACE who introduced its application to a broad audience. (USACE, 1989 - USACE: Expected annual flood damage computation [available online at https://www.hec.usace.army.mil/publications/ComputerProgramDocumentation/CPD-30.pdf] last access 2-5-21, 1989. A source will be added also for the critical need of granting access to WSS main plants for repairs and replacement during flooding.

202: instead of the tiny URL, provide a proper source description. What type of railway is this etc.

Reply. A proper citation to "Technical overview: Payments relating to disruption" of Network Rail (UK national railways) will be specified.

302: sources for the quantitative measures? or is this computed by you? until 320: it seems you have computed those values; what data did you use to achieve it (i.e. road network data sources, types etc.?)

Reply. Authors made the simulations and obtained the results described in the section. For the Water Supply System, the data about the piping network (e.g. pipe diameter, nodes, demand at nodes, position and capacity of storage tanks, position and power of the pumps and lifting station) were provided by Publiacqua s.p.a. the society in charge of managing the integrated water cycle in the area. The road network information (road shapefile and associated attributes related to type, speed, etc.) are openly available in the geographic data portal of the Region. For both models, the exposure analysis was carried out by intersecting the network with flood hazard maps as described in the "case study" paragraph. We will better describe the WSS data and road network data necessary for running the model in section 2.

Table2 very nice and relevant results. Could you provide estimates of possible error margins? A sentence or two would suffice, maybe in the discussion. Just since this looks too exact.

Reply. The results are integer numbers because they are calculated as the sum of the population equivalent (PE) assigned to each affected node, i.e. a node not capable of supplying the nominal demand. PE is based on the density of resident population (census data) and type of the demand (e.g. residential, industrial etc.). Errors in the estimation of PE are comparable to the variations of resident population in census data (updated every 10 years) and of the order of a few, negligible percent in this analysis. We will add this information in the discussion.

356 check grammar

Reply. Grammar will be checked.

359 do those SUVs permit higher wading depths, such as trucks (0,5m)?

Reply. The instability threshold for any vehicle in floodwaters is function of both: (i) water depth and velocity and (ii) geometry and weight of the car (see a whole paper by the first author devoted to this hydrodynamic analysis: Arrighi et al. 2015, "Drag and

lift contribution to the incipient motion of partly submerged flooded vehicles", Journal of fluids and structures). In principle, SUVs and other emergency vehicles are less vulnerable than passenger vehicles (higher planform height and heavier); however, since the flow velocity is not known in this application and the lifting station is very close to the river banks, 0.3 m are assumed as a conservative threshold.

399 This must not be so, road nodes and section, crossing could also be of interest. Maybe just add "within this study"

Reply. The text will be amended.

403 add a source for Value of Time, or VOLL

Reply. A source for the Value of Time will be added (e.g. Pregnolato et al., 2016)

428-433 Interesting areas for future work. Might be illustrative for readers to add a few sources as examples who covers bridges, power supply, operation times etc.

Reply. We will add some new references for the suggested purpose.

Literature Some sources cited in the text are missing in the reference list.

Reply. The reference list will be checked.

---

## Author Comment (AC2) · 12 Apr 2021

In this paper, the authors developed a risk analysis of the water distribution system (WSS) and the road network system under flooding events. The case study is the metropolitan area of Florence, which is in a flood-prone area. The paper aims to study the interdependence between the WSS and the road system by evaluating the accessibility to critical components of the WSS. Network models and topological metrics (e.g. the length of the disrupted edges, network service areas) are used to measure the vulnerability of the systems. Overall, I found this paper interesting and relevant to the field of infrastructure resilience. In particular, the paper tries to analyse interconnections between two infrastructure systems by looking at a real-world case study.

Anyway, the paper still needs some work to be ready for publication. Therefore, I hope my comments will help in the revision process.

Reply. Authors thank the reviewers for the comments. The manuscript aims at highlighting how crucial is the understanding of indirect impacts of flooding to critical infrastructures especially where interdependencies exist. A significant effort was put on identifying metrics that could be suitable for both the analysed networks, from an engineering perspective (in terms of 'disrupted' lengths) and from a societal perspective (people cut-off a service). We also show that a critical component of the WSS network becomes even more critical when looking at interdependencies with the road network because of the difficult accessibility. We think our point-by-point reply (and relative edits) will address concerns and improve the manuscript for publication.

- The syntax of the whole text should be revised. In particular, I found sections 2 and 3.1 difficult to read. Moreover, I found many typo errors in the text (for example, line 234: (2-3"), line 388: "be be").

Reply. We will check Sections 2 and 3.1 and rephrase these paragraphs. We will also correct typos.

- In line 37, you presented previously published works and you wrote "Among these works, indirect impacts and cascade effects are mostly addressed with complex conceptual frameworks (Fekete, 2019; Emanuelsson et al., 2014), :" How do you define a "complex conceptual framework"? Please, be more precise when reporting other works.

Reply. We meant that the conceptual framework cited would require a significant number of models and data to describe the complexity of the systems. We will rephrase the citation.

- Line 50: "Modern cities are currently defined as "systems of systems", where the "systems" are Critical Infrastructure (CI) systems (Gardner, 2016)." I do not agree with

this definition. For me, a city is made also of people, cultures, the environment, the ecosystem, etc. Falco published a paper about this (Falco (2015) "City Resilience through Data Analytics: A Human-Centric Approach).

Reply. We opted to cite a definition of Critical Infrastructure from the engineering and urban science literature (Gardner, 2016). Authors are aware how relevant is the societal sphere (or system indeed); however, this study's niece area is within engineering and urban science, thus other perspective are out of the scope. We fully understand that considering the human sphere is key in understanding indirect (often intangible) impacts and we tried to go in that direction. In fact, we presented different impact metrics where the people are central by evaluating the population not served and delayed commuters. We will broaden the definition of cities in the paragraph by adding the societal components and the suggested references.

- In the text, you wrote often about the "impedance time". For example, in line 372: "Service Areas (SAs) are applied to understand which portions of the city are accessible within a given time, i.e. the impedance time." I have never heard about it. I checked on a vocabulary and it says it is related to electronic measures. Therefore, I am not sure if it is the most correct terminology.

Reply. The concept of Services Areas is well-known in transport and network analysis (here, a quick overview https://desktop.arcgis.com/en/arcmap/latest/extensions/network-analyst/service-area.htm). We appreciate that the multi-disciplinary audience of this article could not grasp the concept, therefore various references will be added and the concept clarified.

- I found often "silo-based" in the text, but there is not a clear definition of it. I think it is important to add a definition because "silo-based" is a relevant concept for your work.

Reply. Also the concept of "silo" is quite common in network and networked infrastructure studies, and it used as the contrary of "system". A silo-based approach (or

thinking) does not consider the relations among different elements (or components, groups, etc.), as opposed to a system-based approach that analyses the interactions among elements. This will be clarified in the text when "silo" appears for the first time.

- Line 58: "Therefore, CIs cannot be considered independently, and silo-based analyses are completely inadequate to understand the behaviour of a given infrastructure operating in its environment (Dueñas-Osorio et al., 2007; Rinaldi et al., 2001)." This statement sounds a bit strong. I checked the two papers. Duenas-Osorio et al. (2007) wrote in their abstract that "Effective mitigation actions could take advantage of the same network interconnectedness that facilitates cascading failures", while Rinaldi et al. (2001) wrote "When examining the more general case of multiple infrastructures connected as a "system of systems," we must consider interdependencies." You wrote in line 83: "Silo-infrastructure studies are limited in their scope since they ignore cascade effects and thus underestimate impact (De Bruijn et al. 2019)." I think that analyses of a single network system can advance our understanding of specific systems or they can help to find metrics to use for other analyses. In section 3.1, you also analysed the WSS and the road network separately. Based on those results you could measure the Annual Average Loss (AAL) on page 12. Overall, I think that the introduction should be revised from this perspective.

Reply. The analysis of a single network can advance the understanding of impacts; however, when the networks have interdependencies, cascade effects amplify and multiply the impacts. Silo-based studies can partly describe flood consequences and in the authors' opinion dependencies among infrastructures should be carefully analysed. We will amend the introduction to better describe the existence of cascade effects within a single network and the amplification effect due to interdependencies.

- In this paper, you used network models embedded in space for your analyses. Anyway, the paper did not report enough literature about this topic. Moreover, other papers studied the impact of floodings on road networks. For example, Casali and Heinimann (2019) "A topological characterization of flooding impacts on the Zurich road network.

PLoS ONE 14(7)"; Kermanshah and Derrible (2017) "Robustness of road systems to extreme flooding: using elements of GIS, travel demand, and network science." Natural Hazards, 86.

Reply. We will add the two suggested references, and review the literature to briefly mention network models. Our literature review focused on flood indirect impacts and cascade effects evaluation on multiple (interdependent) infrastructures. We are aware that the topic is complex and could embrace an even more comprehensive literature review (inclusive of e.g. flood risk analysis, vulnerability of linear infrastructures, indirect impacts and cascade effects, road transport models, WSS network models, resilience of infrastructures); however, this type of revision would be a review paper itself, rather than the contained background of our piece of work.

- Line 120: "Very few studies (Pant et al., 2018; Dong et al., 2019) developed a truly holistic application to analyse interdependency effects; however, indirect consequences are not investigated, especially regarding the WSS-roads interaction." I found this sentence too strong. I would rephrase it because there are many published works that analyzed cascading effects on networks. Moreover, you reported the work of Dong et al. (2019), who developed percolation analyses on the road networks of different cities, not of interconnected networks. Therefore, why is a percolation analysis a truly holistic application for interdependency? I think that even the study of a single network system can represent a holistic approach since it looks at the network system as a whole. Casali and Heinimann developed a thesis from this concept (Casali (2020), "Topological Assessment of Changes in Road Network Systems in Time, under Discrete Flooding Events, and under Classes of Unexpected Disruptions").

Reply. We will rephrase the sentence. We agree that various existing works analyzed cascading effects on networks. However, no study (at our best knowledge) has so far addressed the quantification of indirect impacts and cascade effects in relation to WSS-roads interaction. Our vision of "holistic approach" includes complexity, which is based on interconnections (which cannot be represented by the analysis of a single

system) since a wider 'system of systems' should be considered in the analysis. This vision may differ from other interpretation of "holistic approach", e.g. the analysis of a single network using multiple disciplines. The text will be amended to include both interpretations.

- Why do you use to measure indirect impacts: (i) the length of the disrupted network; and (ii) the population which experiences loss of service? Maybe in section 2.2 you can add more text about the motivations. Moreover, why is the total length of edges a better metric to analyse network vulnerability than other metrics (for example, the number of disrupted edges)?

Reply. Authors aimed at calculating two different types of impact metrics with different perspective: (i) an engineering measure of total length, which can be used as a proxy of potential damage/recovery costs (e.g. the cost of flushing a contaminated pipe with disinfectants is expressed in €km of pipe); (ii) a people-centered measure which describes the amount of population experiencing the interruption/delay of service. This choice is also the result of an effort in identifying metrics suitable for both networks. Moreover, the number of disrupted edges does not provide a quantitative measure since single edges can be 100 or 1000 meters long especially when the study area spans in urban, suburban and rural areas and includes a varying density of infrastructures.

- In the methodology section, I found that not all the information is reported fully. I do not find a definition for the Annual Average Loss (AAL) and details about how you calculated the PPH. Moreover, you introduced the Pressure Driven Demand (PDD), and it can be useful if you will add more information about it.

Reply. AAL is the frequency-weighted amount of loss that can occur on average in any year, we will add a definition and reference to a widely used manual by USACE who introduced its application to a broad audience. (USACE, 1989 - USACE: Expected annual flood damage computation [available online at

https://www.hec.usace.army.mil/publications/ComputerProgramDocumentation/CPD-30.pdf] last access 2-5-21, 1989. In the design of WSS networks the demands can be assumed as defined input data since the main objective is simulating correctly operated networks, this is the standard approach in modelling codes such as EPANET. However, when simulating strongly off-design networks, nodes featuring a reduced pressure are quite common, so that a PDD approach is needed (Arrighi et al., 2017). In PDD models nodal demands are not attributed a priori; instead, their value depends on the current local pressure. In particular, and consistently with practice, the model assumes that each node is in one of three states: fully-served (pressure equal or higher than nominal pressure, partially-served (positive pressure, but lower than nominal pressure), not-served (zero pressure) (see further details in Arrighi et al., 2017). We will add further explanation on this point in the text. PPH (People per Hour) was defined in Sec 2.2 and the definition will be improved.

- In the methodology section, I did not find precise information about how you modelled the networks. For example, what are exactly a node and an edge in the WSS and the road networks? Which software did you use to model them? I understood that the road network extended to a larger area than the WSS network, is that correct? Did the road network add some weights to the edges? For example, in line 217, you wrote that "for the flooded scenarios, the network properties of a link (i.e. travelling speed) are modified according to the functions, and traffic parameters recalculated for the perturbed state." This means that you used the travelling speed in the analyses of the road network. Therefore, how did you calculate the travelling speed?

Reply. In the WSS model the network is composed by edges, i.e. pipes with assigned diameter that transport potable water from the treatment plant to the users, which intersect each other at nodes of known elevation (below the ground) and where a nominal demand is assigned depending on PE. The network is modelled with EPANET (see section 2.2) with a user modified PDD calculation scheme (see reply to previous comment). The analyzed road network extends to a larger area because indirect impacts

to commuters extend beyond the municipalities served by the WSS. Regarding the road network, the perturbed travelling speed of flooded roads was computed using the flood-transport function of Pregnolato et al. (2017b), as mentioned. We agree this was poorly explained and will be improved.

- You used the SA (network service area) to look at accessible areas. Did you consider also directions of roads when you analyse the shortest paths?

Reply. As reviewers have appreciated, this study is highly complex and multi-disciplinary; moreover, it is based on publicly available data. Therefore, some modelling simplifications were necessary. Network SAs used regional cartography road data, which lacks complete information, and applied through an ArcGIS model, which unfortunately could not create road directions. Improving the sophistication of the model is definitely in the future development of this piece of research; this will be added in the appropriate section.

- Line 226: " The widely accepted definition of resilience is the ability to overcome an impactful event and return to normal condition through a quick recovery;" There are many authors that defined resilience in recent years. You can add a reference to a published work on infrastructure resilience.

Reply. We will add a reference on the definition of resilience which is indeed defined in several ways according to literature, for example the recent review by McClymont et al. (2020): McClymont et al. (2020), Flood resilience: a systematic review, Journal of Environmental Planning and Management, 63:7, 1151-1176, DOI: 10.1080/09640568.2019.1641474

- Figure 4: you can improve the resolution of the figure. Moreover, I cannot see the edges of the urban network in figure a.

Reply. Full resolution figures will be submitted with the revised manuscript. The urban network is extremely dense in the area so a light grey color was selected, but we will

try to find a better compromise to obtain a clearer visualization.

- Figure 5: I cannot read legend of figure b. The description of figure c is missing. You can improve the resolution of this figure. Moreover, what does it mean "low", "medium" and "high" in legend of figure c?

Reply. The legend will be improved, and the description of panel c will be added. The meaning of "high" is that the municipality is affected by a loss of functionality of the unique source of water supply, "medium" is when a municipality might rely upon an alternative water source but still will experience some loss of service; "low" is when the municipality does not experience loss of service (see description of the municipal WSS connection in sect.2 ll. 277-284). This more detailed description will be added to the text.

- Section 3.1 "Silo base analyses": you can add a topic sentence to introduce the "silo-based analyses".

Reply. The concept of "silo" has been clarified in the text when "silo" appears for the first time (L59), as suggested in previous comments.

- Caption Table 2: I would repeat here that the WSS is not affected for 30 and 100 years events.

Reply. We will modify the caption accordingly.

- Figure 6: it is not clear how did you choose the interval limits for this figure because it is not reported in the method section.

Reply. The detail about how the interval limits were defined are not present because we did not think it was relevant; in fact, the classification of Fig. 6 is not used throughout the text but for visualization only (and of Fig. 6 only). The actual numbers of PPH were used for analysis and drawing conclusions; nevertheless, we thought that a qualitative representation of the distribution of PPH could have been a nice (graphic) addition to the paper.

- Table 3: why there is not Florence in this table? Then, I would order the municipalities as in table 2.

Reply. Table 3 shows the impact on the commuting population to Florence. Therefore, Florence is not represented because internal commuting within Florence is not considered (due to lack of data). Tables will be both ordered by magnitude: Table 2 will rank elements using PE (the first line will have the highest PE), while Table 3 using PPH (the first line has the highest PPH). We think this visualization has a good rationale, and we will amend Table 2 with this approach.

- Lifting stations are important for the analyses of this paper. You can describe more the geography of lifting stations, for example, how many and where are lifting stations? Therefore, I ask the authors to provide more description of the topology of the WSS in the result section.

Reply. The main lifting station of the WSS system is located downstream of the treatment plant of the city as depicted in Fig. 4 with orange triangle symbols. This lifting station is located very close to the river and it is affected for medium-low recurrence intervals event with the effects described in the whole manuscript. The water undergoes treatment and reaches the lifting station, where six 710kW pumps ensure a maximum head of 60m and feed the distribution network. The 17 storage tanks are mostly located at high altitudes and feature a total operative volume of 48 620m3. The storage tanks are equipped with smaller pumps to ensure their day-night operativity. In case of power shutdown, the transient behaviour of the system is determined by the amount of water stored in tanks. In a previous work (Arrighi et al., 2017) a sensitivity analysis of the WSS behavior with respect to the tanks level has identified two conditions. Water levels variations in low-altitude tanks strongly impact most network nodes; high-altitude tanks, have a smaller area of influence limited to the immediate surroundings of the tank itself. We will add a better description of the topology as suggested.

- In Line 389 you wrote, "The first recommendation is then to develop ad-hoc emergency plans by identifying potential critical hotspots (e.g. WSS lifting stations), and to analyse the accessibility to these sites." Therefore, you concluded that identifying critical components in a single network can help to improve emergency plans. Is my interpretation correct?

Reply. We think both, i.e. identifying critical hotspots in one network is important (this was done by previous studies) and in particular the identification of those hotspots where the cascading effects is more relevant (and here we can refer back to our case study). The identification of critical components in a single network is helpful, but preparedness plans should also consider how the network interacts with other infrastructures. In our case study we demonstrate that if the lifting station is affected, one can assume that an immediate assistance can be provided to restart the pumps with limited effects on the water supply. However, this is not possible if the area is not accessible and the consequent 'unexpected' delay further aggravates the impacts (see Fig. 5, b). In conclusion, a critical component of the WSS is not just a node/section that can be affected causing wide effects on the network itself, but also difficult to repair because of the impacts on another interconnected network, i.e. the road network.

- Line 393: "The third recommendation is to enhance the system redundancy for those municipalities totally reliant on a single main system, with e.g. emergency water storage tanks." I agree but by adding new infrastructures maybe we add new externalities to the urban area? What are the immaterial costs of building new water storage tanks in Florence? This paper might be interesting Chelleri and Baravikovab (2021) "Understandings of urban resilience meanings and principles across Europe".

Reply. We agree with this observation and we will be more cautious with the recommendation, i.e. by saying to evaluate increasing the redundancy with respect to considering other factors such as costs, environmental impact in a sustainable perspective.

- Conclusion part, line 444: "Results showed that the impact of flooding to the two systems differs in both spatial (up to 5 affected municipalities per WSS, 37 for the

road system) and temporal scale (60 minutes before first pressure drop, 30 minutes to reach critical depths on roads)." I cannot find in the result part where you reported or discussed the 30 minutes to reach the critical depths on roads. Please, write this result in the main result section.

Reply. We will amend the text to keep results in the main result section.